# Associations between Dietary Intake, Blood Levels of Omega-3 and Omega-6 Fatty Acids and Reading Abilities in Children

**DOI:** 10.3390/biom13020368

**Published:** 2023-02-15

**Authors:** Francesca Borasio, Valentina De Cosmi, Veronica D’Oria, Silvia Scaglioni, Marie-Louise Eva Syren, Stefano Turolo, Carlo Agostoni, Marilena Coniglio, Massimo Molteni, Alessandro Antonietti, Maria Luisa Lorusso

**Affiliations:** 1Scientific Institute IRCSS E. Medea, Unit of Child Psychopathology, 23842 Bosisio Parini, Italy; 2Department of Psychology, Catholic University of the Sacred Heart, 20123 Milan, Italy; 3Department of Clinical and Community Sciences, University of Milan, 20122 Milan, Italy; 4Fondazione IRCCS Cà Granda Ospedale Maggiore Policlinico, Anestesia e Terapia Intensiva Donna-Bambino, 20122 Milan, Italy; 5Fondazione De Marchi, Department of Pediatrics, Fondazione IRCCS Ca’ Granda Ospedale Maggiore Policlinico, 20122 Milan, Italy; 6Fondazione IRCCS Ca’ Granda Ospedale Maggiore Policlinico, SC Nephrology Dialysis and Pediatric Transplantation, 20122 Milan, Italy; 7SC Pediatria-Immunoreumatologia, Fondazione IRCCS Ca’ Granda Ospedale Maggiore Policlinico, 20122 Milan, Italy; 8Fondazione IRCCS Ca’ Granda Ospedale Maggiore Policlinico, SC Child and Adolescent Neuropsychiatry, 20122 Milan, Italy

**Keywords:** PUFAs, reading, writing, phonological awareness, dyslexia, dietary intake, omega-3, omega-6, omega-6/omega-3 ratio, blood PUFAs

## Abstract

Lower levels of omega-3 polyunsaturated fatty acids (PUFAs) have been described in individuals with reading difficulties, but the degree and the nature of such deficiencies as well as the role of nutrition are a matter of debate. The aim of the present study was to investigate the associations between PUFA blood levels, nutritional status, and reading/writing/phonological awareness performances in 42 school-age children with varying levels of reading ability. Significant correlations were found between PUFA levels (specific omega-6/omega-3 ratios), the ratio of omega-6-derived calories to the total amount of calories and reading scores. Mediation analysis showed a mediating effect of fatty acids on the association between reading speed scores and nutritional status. Moderation analysis, moreover, showed that the associations of omega-6/omega-3 ratios in the blood and Kcal omega-6/Kcal total in dietary intake were moderated by reading speed performances. Results of the mediation and moderation models confirm that the associations of dietary intake with PUFA levels in the blood vary depending on learning abilities. Reading skills appear to be sensitive to the effects of a complex set of favorable conditions related to the presence of higher omega-3 blood levels. These conditions may reflect the action of dietary as well as genetic and epigenetic mechanisms.

## 1. Introduction

In recent years, several studies focused on the potential role of polyunsaturated fatty acids (PUFA) in different neurodevelopmental, psychiatric, and neurodegenerative disorders. Polyunsaturated fatty acids (omega-6 and omega-3) play an important role in the proper anatomo-functional development of the brain [1,2]. They are involved in the maturation and functioning of the neurons, fluidity of the plasma membrane, and gene expression, and are critical elements for cell transduction and learning processes [3,4]. Some studies suggested that abnormalities of fatty acid metabolism may contribute to different neurodevelopmental disorders, including ADHD (attention-deficit hyper-activity disorder) and Developmental Dyslexia (DD) [5,6,7,8,9].

DD is one of the most common learning disabilities. It is defined as a specific reading disorder despite normal intelligence and the absence of sensory or neurological deficiencies [10,11]. DD is a heterogeneous disorder: Although phonological deficits have been described as one of the most distinctive features of children with DD [12,13,14], many researchers agree with the idea that several other factors contribute to explaining the emergence of reading disorders (e.g., [15,16]). Besides phonological processing [12,13], low-level visual and auditory perception and attention mechanisms [17,18,19,20], as well as memory and executive functions [12,15,16], have been called into play and fatty acid metabolism seems to be one of the possible underpinning factors [21].

Of the total lipid content of the brain, 35–40% are PUFAs, such as eicosapentaenoic acid (20:5n3; EPA), docosahexaenoic acid (22:6n3; DHA), and arachidonic acid (20:4n6; AA). In vertebrates, DHA is the major PUFA of the brain together with AA [22]. Linoleic acid (18:2n6; LA) and alpha-linolenic acid (18:3n3; ALA) are classified as essential fatty acids (EFAs), indicating that they play a crucial role in human metabolism, but they need to be acquired through nutrition since they cannot be synthesized from other nutrients [23]. This incapacity is due to the lack of two genes for the enzymes acting on the omega-6 (Delta-12 desaturase absent) and omega-3 (Delta-15 desaturase absent) positions. As a consequence, the two EFAs have to be present in the diet [24]. Thus, the body synthesizes AA (along with other omega-6 PUFAs) from LA and EPA and DHA (along with other omega-3 PUFAs) from ALA [23,25]. PUFAs are found in high proportions in plant-based foods and humans have the capacity to metabolize EFAs into long-chain derivatives [26]. The omega-6 to omega-3 ratio is also very important to human health [27]: an overabundance of fatty acids from one family will limit the metabolic production of the longer chain products of the other one.

Several national health agencies recommend an intake of omega-6 to omega-3 ratio of approximately 4:1 [27]. However, the typical Western diet is characterized by a higher intake of LA compared to ALA (by five to 15 times higher), in a ratio ranging from 8:1 to 25:1 [28]. Studies showed that an optimal ratio of omega-6 to omega-3 positively affects biological processes such as inflammation, although their role in the context of inflammation is not yet clear [26]. Omega-3 PUFAs possess antioxidant and anti-inflammatory properties [29,30]. Omega-6 is mainly pro-inflammatory in character. It has been suggested that the ratio AA/EPA is the best upstream marker of cellular inflammation [31]. However, at the current intake levels of LA and AA, there seems to be a limited effect on inflammatory biomarkers [26].

Considering neurodevelopmental disorders, and DD in particular, some studies found an association between severe fatty acid deficiencies and poor reading, spelling, and auditory working memory [8]. The association of long-chain PUFAs (LCPUFAs) with learning, visual memory, and information processing speed [32,33,34], moreover, seems to be found regardless of the diagnosis of DD. A recent study by Borasio et al. [35] showed a direct relationship between PUFAs and reading/writing skills, ranging over the whole spectrum of reading abilities.

As previously said, essential fatty acids cannot be synthesized by the organism but must be obtained from dietary sources [4]. The predominant sources of omega-3 fatty acids are vegetable oils and fish. Specifically, ALA is found in the chloroplast of green leafy vegetables (e.g., spinach) and in seeds of flax, linseed, and walnuts, while fish is the main source of EPA and DHA. Concerning omega-6, vegetables are the main sources and LA could be found in different kinds of vegetable oils such as corn oil, safflower oil, sunflower oil, and soybean oil [36].

Since the prenatal period, low maternal intake of omega-3 during pregnancy has been shown to be associated with different neurodevelopmental disorders, such as ADHD, DD, dyspraxia, and autism [8,9,37]. There is currently no clear recommended dietary allowance (RDA) for omega-3 PUFAs for the entire lifespan, including pregnancy. The increased need during pregnancy is due to the fetus requiring higher amounts of omega-3 PUFAs, which appear to be crucial for the formation of cell membranes in some tissues, including brain tissue, and the development of neurons and synapses [38]. Indeed, the developing brain has been shown to be more susceptible to omega-3 PUFA deficiency than the mature brain [38].

The brain continues to develop also in childhood and adolescence and nutrition continues to have an impact on cognitive ability and behavior [39].

DHA has an important role in various brain developmental processes such as neurogenesis, neuritogenesis, migration, synaptogenesis and neurotransmission, learning, and memory consolidation [40,41]. The evolution of higher-order brain functions was attributed to omega-3 intakes and DHA was found to be densely accumulated in brain areas associated with learning and memory [40]. Both ALA (omega-3) and LA (omega-6), moreover, can directly increase the fluidity index of the cell membrane [3]. The child’s cognitive functions related to attention, memory, and verbal scores are involved in the optimal development of the prefrontal cortex [12,17]. This area of the brain may require high amounts of omega-3 PUFAs during development due to the complexity of its synaptic connections [38].

As reported by Sinn, Milte, and Howe [42], omega-3 fatty acids have also been associated with substantially increased transport of glucose across the blood-brain barrier [43,44]. Therefore, it is also possible that their primary influence on brain function includes improved cerebral blood flow and blood-brain barrier integrity [45].

As previously said, modern diets lack omega-3 fatty acids intake, with possible consequences on children’s learning, reading, and attention abilities [46]. It has been shown that the negative effects of early depletion in DHA intake on brain functions are difficult—but not impossible—to reverse [47], and also that high levels of omega-6 fatty acids intake may contribute to the reduction of DHA in the brain [48].

Associations between PUFA dietary intake and PUFA levels in the blood could be, however, variable. A recent work by Mulder et al. [49] showed a high degree of variability in school-aged children, with some children with low intakes of DHA having a blood concentration equivalent to some children with high intakes. Mulder and colleagues also found different associations between DHA intake, DHA blood level, and some neurodevelopmental measures: DHA in the blood but not dietary DHA was associated with multiple tests of cognitive performance. The results of the study and the variability of the association between dietary intake and PUFA blood level raise complex questions on the relation between diet, PUFA transfer to membrane lipids, and neural functioning. Baylin and colleagues [50] stated that fasting whole blood could be the sample of choice in epidemiologic studies because of its ability to predict intake and its accessibility. Regarding neurodevelopmental disorders, fatty acid profiles seem to be altered amongst children with ADHD and autism spectrum disorders, with low EPA and DHA levels and a high ratio of omega-6 to omega-3 fatty acids [51]. Indeed, children with specific learning disorders are often reported to have an imbalance or deficiency in fatty acids (see the case report by Baker, 1985) [52]. Beyond dietary factors, constitutional factors could be potential modulators of omega-3 status: “conversion” efficiency of fatty acid precursors to their long-chain derivatives, including DHA and EPA, has been shown to be associated with genetic variability in the fatty acid desaturase (FADS) gene cluster, coding for enzymes that metabolize omega-3 PUFAs [53]. Two studies suggested that individuals with DD are more likely to have a fatty acid deficiency, based on specific signs (FADS) that include excessive thirst, frequent urination, dry skin, dry hair, brittle nails, dandruff, and follicular keratosis [8,54]. Kirby [55] measured cheek cell omega-3 fatty acid levels in school children but found no correlations between omega-3 fatty acid levels and reading or spelling abilities. Within children with DD, high levels of omega-6 and low levels of omega-3, or high omega-6 to omega-3 ratios, were found to be associated with worse performance in reading and writing [8,35]. The extent to which PUFA imbalances in children with DD could be related to lack of intake and/or to other genetic and environmental factors has still to be established.

Studies found that consumption of omega-3, particularly DHA, by school-aged children may enhance cognitive performance, especially in those who habitually consume diets low in omega-3 [8,46,55,56]. The improvements concern cognitive outcomes, such as memory, processing speed, visual-perceptual ability, attention, reading, and spelling skills. A large German study [57] on the impact of fish consumption on school performance in children found a positive association involving both math and German language grades, which was not linear but tended to decrease again in the highest categories of fish intake (possibly due to detrimental effects of mercury or other pollutants in the fish at high intake levels). The key role of omega-3 LCPUFA has been specifically associated with DHA as it is a key component of membrane structural lipids, especially in nervous tissue and the retina. The developing brain accumulates large amounts of DHA both before and after birth, particularly during the first two years of life with DHA predominantly acquired from the mother through placental transfer and breast milk [58]. The brain’s ability to synthesize DHA then increases with gestational age [58]. The sum of DHA and EPA over the total amount of fatty acids in erythrocytes (known as the “Omega-3 index” [59]) has been recently described as a very useful indicator of cardiovascular health, but no data have been reported to date that suggests a role of this index in predicting cognitive development and functions.

Based on previous research that found different associations between PUFA dietary intake, PUFA levels in the blood, and neurocognitive measures [49], the aim of the present study was to investigate the correlations between PUFA blood levels, nutritional status, and reading/writing/phonological awareness performance. Specifically, the first aim was to test the hypothesis that the effect of nutrition on learning skills is mediated by PUFA levels in the blood. Our second hypothesis was that children with lower reading abilities may be less able to process and deposit dietary PUFAs in brain tissues, which implies that the association of nutritional intake and the actual PUFA levels in the blood would be moderated by performance on reading, writing, and phonological awareness tasks. In other terms, according to this hypothesis, the association between nutrition and blood levels of PUFAs would be expected to differ at different levels of learning abilities. This could reflect the effects of other intervening variables linked to the genetic and constitutional differences that lie on the basis of the learning ability itself.

## 2. Materials and Methods

### 2.1. Participants

A total of 42 Italian students (including 25 children with DD and 17 normally-reading children) took part in the present study. Participants with reading disorders were selected among patients of the neuropsychiatry unit of IRCCS “Eugenio Medea” in Bosisio Parini, Northern Italy, while typically developing children (TD) were recruited in the same geographic area among the schoolmates and friends of the children with DD. Participants were recruited between July 2020 and September 2022.

Participants had to fulfill the following inclusion criteria: (a) age between 7 and 15 and attending at least the third grade of primary school; (b) IQ ≥ 80; (c) monolingual speakers or bilingual speakers with perfect (native-like) mastery of the Italian language. Inclusion criteria for children with DD were: (a) having been previously diagnosed with DD based on standard inclusion/exclusion criteria [11]; (b) absence of comorbidity with ADHD and other neuropsychiatric or psychopathological conditions (comorbidity with other learning disorders was allowed); (c) not having received neuropsychological treatment for DD before. Criteria for TD children were: (a) normal school achievement as reported by teachers and parents; (b) no z-scores below −1.5 with respect to age mean in word and nonword reading tests and in tests of writing to dictation (DDE-2 battery) [60].

The children and their families were contacted by the researcher and written parental informed consent was obtained before participation in the study. The study was approved by the Local Ethics Committee in accordance with the Declaration of Helsinki.

### 2.2. Measures and Procedure

The present study is part of an ongoing project on the efficacy of PUFA supplementation to enhance the effects of neuropsychological treatment for children diagnosed with developmental dyslexia (registered in ClinicalTrials.gov, Code NCT04287530).

All measures involved in the present study were collected as part of the pre-test assessment of the main project. Reading, writing, and phonological awareness tests were administered individually by trained psychologists in one session of 90 min, approximately. The blood samples were collected by nurses of the Scientific Institute Eugenio Medea during the same assessment session. Lastly, the Food Frequency Questionnaire (FFQ) was administered remotely to each child and parent by trained nutritionists of the Pediatric Intermediate Care Unit of IRCCS “Ca’ Granda Ospedale Maggiore Policlinico” in Milan, Italy.

#### 2.2.1. Blood Measurements of Fatty Acids

The analysis of PUFA blood levels was based on the methodology described by Marangoni et al. [61] and Borasio et al. [35]. Blood samples were obtained by collecting a drop of blood from a fingertip. Samples were collected on Whatman 903 collection cards BHT pre-treated and stored at a temperature of −20 °C. The dried blood spot was methylated with HCl/MeOH (Supelco, MERCK), and the fatty acid methyl esters (FAME) were extracted with hexane and injected into Shimadzu Nexis GC-2030 gas chromatograph. A 30 m capillary column (FAMEWAX, RESTEK) was used to separate the FAME. The Labsolution software v.5.97 SP1 (Shimadzu) was used to identify FA species using the retention time of standards (PUFA1, PUFA2, PUFA3 (Supelco, MERCK) and NHI-F (AccuStandard, RESTEK).

Total omega-3 and DHA compositions were measured as a percentage of the total fatty acid. Moreover, the ratios omega-6/omega-3, AA/EPA, AA/ALA, and LA/ALA were computed. These were considered as the measures of interest based on previous literature. Other indexes were not taken into consideration.

#### 2.2.2. Nutritional Status and Food Frequency Questionnaire (FFQ)

Body weight and body length were reported by parents. Body mass index (BMI) was calculated as weight (kg)/length or height (m2). Z-scores for weight for age, BMI, and weight for length were calculated using the World Health Organization (WHO) Anthro and Anthro Plus^®^ software [62,63] and the WHO reference charts [64]. Nutritional status was classified according to WHO criteria [65].

Dietary habits were assessed by means of an age-adjusted Food Frequency Questionnaire (FFQ) made up of 99 items and designed according to Block et al. [66] and Bellù et al. [67]. In a validation study, with a standard measurement error model, correlations for energy between estimated truth and the Block FFQ were 0.45 for both women and men [68]. A trained dietician interviewed mothers for approximately 40 min and each meal was analyzed to determine which food was eaten and how often. Quantification and analysis of energy intake and nutrient composition were performed by an ad hoc PC software program (Metadieta software, Rome, Italy). The daily amount of proteins, total fats, available carbohydrates, soluble carbohydrates, and fibers, fatty acids consumption (overall and as MUFA, PUFA, and SFA) was estimated using the most recent update of an Italian food composition database (Food Composition Database for Epidemiological Studies in Italy by Gnagnarella et al. [69]). Consumed foods were traced and data were extracted from a validated modified KIDMED questionnaire [70]. The score obtained reflects adherence to the Mediterranean Diet (MedDiet) in that a low score indicates low adherence to the MedDiet and a high score indicates high adherence [70,71]. Values ≤ 2, 3–6, and ≥7 were considered low, intermediate, and high levels of adherence to MedDiet, respectively. Based on previous literature, the following indexes were considered as variables of interest for the present study: total energy intake, total macronutrient intakes, particularly, the intakes of carbohydrates, total lipids, total PUFAs and the ratios kcal n-6/kcal tot (%) and kcal n-3/kcal tot (%). The Italian Guidelines set the reference intake for omega-6 PUFAs as 4-8% of the total energy intake for children and adolescents and set the omega-3 PUFAs reference intake as 0.5-2% of the total energy intake. For this reason, we decided to implement the variables kcal n-6/kcal tot (%) and kcal n-3/kcal tot (%), too. [50,72].

#### 2.2.3. Neuropsychological Tests

The tests commonly employed in the assessment of reading disorders in Italy were used. The results of all measures are expressed as raw scores and converted into z-scores according to the national age norms reported in the test manuals. The following tests were administered:(a)Cognitive measures. The Wechsler Intelligence Scale for Children, Fourth Edition (WISC-IV) [73] and the Raven’s Coloured Progressive Matrices Test (CPM) [74,75] were adopted to assess inclusion criteria related to normal intelligence. The WISC-IV was adopted in the clinical assessment for children diagnosed with DD, while the CPM test was used for typically developing children.(b)Reading and writing tests.


Single word/nonword reading. “DDE-2: Batteria per la Valutazione della Dislessia e Disortografia Evolutiva-2” (Assessment battery for Developmental Reading and Spelling Disorders-2) [60] was used to assess speed and accuracy (expressed in the number of errors) in reading word and nonword lists (112 words and 48 nonwords in total). It provides grade norms from the second to the last grade of junior high school. The test has good validity and reliability scores and is widely adopted by Italian students.Word and sentence writing to dictation. Two dictation tasks were taken from the DDE-2 battery [60], giving accuracy scores (number of errors) according to age norms in writing (48) words and (12) sentences.Phonological awareness. A subtest of the Developmental Neuropsychological Assessment, Second Edition (NEPSY-II) [76] was used to assess phonological processing skills. This test is subdivided into two parts. In the first part of the test, the child is required to establish a correspondence between sounds and images and to select the corresponding image based on a partial sound. The second part of the test requires the child to encode a verbal stimulus and manipulate its phonetic structure by either removing a sound or replacing a sound with another one. Raw scores were calculated as the number of correct answers (range 0–53) and z-scores were computed according to national age norms.


### 2.3. Data Analysis

Data were analyzed with SPSS [77] and Jamovi software [78] on the whole sample (*n* = 42), according to the following steps. In order to reduce the number of analyses and to obtain more reliable scores, three general scores were computed for reading and writing measures based on raw scores: (1) DDE general reading time score, i.e., the average time used for word and nonword reading (expressed in seconds), (2) DDE general reading errors score, i.e., the average number of errors in word and nonword reading, and (3) DDE general writing errors, i.e., the average number of errors in word and sentence dictation. The same process was applied to z-scores, obtaining a general reading speed z-score, a general reading accuracy z-score, and a general writing accuracy z-score. Since z-scores were available only for neuropsychological scores, all subsequent analyses where also dietary data and blood levels of PUFAs involved were conducted on raw scores, partialling out the effect of age. Z-scores, however, were used to compare data from different tests at the cognitive/neuropsychological level.

In order to compute plausible mediation and moderation models, the variables that were significantly associated across levels (nutritional, blood fatty acids, and neuropsychological functions) needed to be identified. Therefore, three types of pairwise comparisons were performed (testing the associations of neuropsychological variables with blood PUFAs, the associations of neuropsychological variables with nutritional variables, and the associations of nutritional variables with blood PUFAs). All correlation analyses were partial correlation analyses on raw scores with age as a control variable. A Bonferroni correction for multiple comparisons was applied in the three steps of partial correlation analysis, taking into account existing correlation patterns within the variables belonging to the same level (for instance, if a given PUFA correlates with reading but reading is highly correlated with writing, it is expected that the same PUFA will also correlate with writing, and the two comparisons will only be counted as a single comparison with respect to Bonferroni correction, see [79,80]). Regarding blood PUFAs, DHA, total omega-3, and omega6/omega3 were considered as one single cluster due to their very high mutual correlation (Pearson’s correlation indexes r > 0.824, all *p*-values < 0.001) and so were the last two indexes (LA/ALA and AA/ALA, r = 0.810, *p* < 0.001). AA/EPA, instead, correlated with total omega-3 and omega-6/omega-3 ratio (r > 0.400, all *p* < 0.009) but not with DHA (r = 0.200, *p* = 0.209), so it was not included in the first index and was considered as a separate cluster. As to the neuropsychological variables, the three reading/writing general scores and the phonological processing score counted as one since they were mutually correlated (rs between 0.576 and 0.722). For the aims of Bonferroni correction, thus, three PUFA clusters were considered to be tested for associations with the neuropsychological cluster (3 clusters × 1 cluster = 3) and alpha was divided by 3 (alpha/3 = 0.017). Regarding nutritional data, three clusters of inter-related variables were identified: total PUFAs, total Kcal, total carbohydrates, and total lipids formed a single, intercorrelated cluster (rs between 0.416 and 0.937, ps < 0.007), while kcal n-6/kcal tot and kcal n-3/kcal tot were not intercorrelated and were considered as two additional clusters. Three comparisons were thus considered for the associations of dietary and neuropsychological variables (3 clusters × 1 cluster = 3, alpha/3 = 0.017). Finally, 12 different tests were considered for the correction of multiple testing in the associations between nutritional data and blood PUFAs (3 clusters × 3 clusters = 9, alpha/9 = 0.006). The correlation matrices of the three levels of variables are reported in Table A1, Table A2 and Table A3 (Appendix A).

After correction with the above-mentioned alpha levels, the triplets of mutually associated variables from the three levels were selected. Standard GLM Mediation models (Jamovi [81]) were adopted to test the possible mediating role of PUFAs in the blood (those found to be significantly associated with reading/writing/phonological awareness performances) in the relationship between nutritional status and reading/writing/phonological awareness. Moreover, moderation models (Jamovi [81]) were adopted to test the possible moderating role of reading/writing ability in the relationship between nutritional variables and PUFA levels in the blood, i.e., to test the possible effect of genetic and other constitutional variables on the absorption of fatty acids from the diet.

Z-scores (with positive performance always expressed by scores in the positive range) were used for reading/writing/phonological awareness measures in all mediation and moderation models, so as to control for obvious effects of age on performance levels. For mediation and moderation models, which were constructed as post-hoc analyses based on the observed correlations, unidirectional hypotheses were considered, and one-tailed significance values were taken (alpha = 0.1).

Power analysis was conducted with G*Power software [82] assuming an average correlation of AA/ALA and LA/ALA with reading/writing/phonological awareness scores between 0.3 and 0.4. The average correlation was based on previous data showing negative correlations of the omega-6/omega-3 ratio with reading and with spelling (0.03 < r < 0.36, average r in the whole sample = 0.24) in a similar sample of children [5], and on data from our own study [35] where average correlations of AA/ALA and LA/ALA with reading, writing and phonological awareness scores ranged between 0.4 and 0.8 (average r = 0.583). First of all, the power analysis conducted for correlations (exact tests, bivariate normal model) showed that 42–43 participants are required to reach a power of 0.80 with an effect size of 0.41–0.42 (two-tailed), corresponding to the average effect size between the two studies (i.e., the average between 0.24 and 0.583). As to mediation, in the absence of previous reference data, a sensitivity analysis (conducted with linear multiple regression, fixed model, single regression coefficient with two predictors, and one-tailed *t*-test) showed that 42 participants would allow detecting a medium mediation effect size of 0.153 with a power of 0.80. Regarding moderation, sensitivity analysis showed that 42 participants would allow for the detection of a moderation effect of 0.197 (linear multiple regression, fixed model, F-test for R^2^ increase with one tested predictor and three total predictors). Even if these figures presuppose medium-to-large effects, they were deemed acceptable considering that the presence of significant and thus rather large correlations would be a necessary, a priori condition for mediation and moderation analyses to be conducted.

## 3. Results

### 3.1. Participants’ Characteristics

The whole sample consisted of 42 children aged between 8 and 13 years (mean age = 10.68 years, SD = 1.39). A total of 25 children were diagnosed with DD (DD) and 17 were normally reading children. At a preliminary check, no differences were found in age and IQ between the two groups of children (all ps > 0.54). The characteristics of the children are presented in Table 1.

### 3.2. Nutritional Status and Food Frequency Questionnaire (FFQ)

The average intake for macronutrients and for SFA, PUFA, MFA, and ratios are presented in Table 2. Regarding adherence to the MedDiet, 20 (47%) children had low adherence, 12 (29%) had medium adherence and 10 (24%) had high adherence. On the whole sample, 12 (29%) children usually ate legumes more than three times/week, 11 (26%) children consumed red or white meat less than two times/week times/week, 40 (95%) children ate meat more than one times/week, and 27 (64%) consumed fish more than three times/week. Moreover, 20 (48%) children usually consumed vegetables almost two times/day and 20 (48%) children consumed fresh fruits almost two times/day, 27 (64%) did not eat nuts/seeds, 40 (95%) children ate sweets almost once a day, 12 (29%) consumed eggs more than two times/week, 37 (88%) children consumed cheese more than two times/week, and four children ate a source of dairy protein (e.g., milk or yogurt once a day).

### 3.3. Correlations between PUFA Levels and Reading, Writing, and Phonological Processing Measures

The relationships between reading/writing/phonological awareness raw scores (general reading time, general reading errors, general writing errors, and phonological awareness accuracy) and PUFA levels in the blood were analyzed through partial correlations with age as a control variable and a Bonferroni correction of alpha/3 = 0.017 (Table 3). General reading time was significantly and positively associated with both AA/ALA and LA/ALA ratios. At post-hoc analyses, AA/ALA and LA/ALA ratios correlated with all the component subtests of the general reading time (r > 0.524, all *p* < 0.001).

### 3.4. Correlations between Nutritional Information and Reading, Writing, and Phonological Processing Measures

Partial correlations controlling for age were computed between reading measures that had been found to correlate with PUFA ratios and nutritional information (Table 4). A Bonferroni correction of alpha/3 = 0.017 was applied. General reading was found to be significantly and positively associated with the Kcal omega-6/Kcal total ratio. Carbohydrates were negatively associated with reading errors.

### 3.5. Correlations between PUFA Levels and Nutritional information

In the last step of the correlation analyses, the associations between PUFA ratios that were found to correlate with reading, measure, and the nutritional status variables that were significantly correlated with learning measures were analyzed (Table 5). For the last partial correlation analyses, a Bonferroni correction of alpha/9 = 0.006 was adopted.

Partial correlations highlighted significant, positive associations of AA/ALA and LA/ALA ratios and Kcal omega-6/Kcal total ratio (all *p* < 0.001).

### 3.6. Mediation Analyses

According to the results obtained with previous partial correlation analyses, two GLM mediation models were tested. General reading speed was used as the dependent variable, Kcal omega-6/Kcal total as an independent variable, and AA/ALA and LA/ALA ratios as mediators. In both mediation models z-scores for reading were used instead of raw scores so as to avoid spurious effects due to age differences.

The first mediation model tested the role of the AA/ALA ratio (mediator) in the association between Kcal omega-6/Kcal total (independent variable) and general reading speed (dependent variable). The indirect effect of Kcal omega-6/Kcal total on reading speed, through the AA/ALA ratio, was significant (*p* = 0.029). The direct effect was not significant (*p* = 0.105). In the component analysis, Kcal omega-6/Kcal total was confirmed to be a significant predictor of the AA/ALA ratio (*p* < 0.001), and AA/ALA was found to predict reading speed (*p* = 0.006) (Figure 1).

The second mediation model tested the role of the LA/ALA ratio (mediator) in the association between Kcal omega-6/Kcal total (independent variable) and general reading speed (dependent variable). In this case, only a direct effect of Kcal omega-6/Kcal total on reading ability emerged (*p* = 0.036), whereas the indirect effect was non-significant (*p* = 0.106). In the component analysis, Kcal omega-6/Kcal total significantly predicts LA/ALA ratio (*p* = 0.002), and also LA/ALA was found to predict reading speed (*p* = 0.060) (Figure 2).

### 3.7. Moderation Analyses

Based on previous partial correlation analyses, two moderation models were tested to assess the moderating effect of general reading speed in the associations of Kcal omega-6/Kcal total (predictor) with AA/ALA and LA/ALA ratios (dependent variables). See Table 6 for the results.

A significant, moderating effect of general reading speed on the relationship of Kcal omega-6/Kcal total and both AA/ALA and LA/ALA ratios (all *p* < 0.001) can be observed.

## 4. Discussion

The aim of the present study was to explore the associations between PUFA levels in the blood, children’s nutritional status, and performance in reading, writing, and phonological tasks. Two hypotheses have led to the investigation. The first one was that the effect of nutrition on learning skills would be mediated by PUFA blood levels. The second one was that the association between diet intake and PUFA could be different at different levels of reading, writing, and phonological awareness abilities, i.e., that it could be moderated by genetic and other neurobiological characteristics of the children, which would be reflected in their learning ability.

The series of correlation analyses conducted to explore the associations of PUFA blood levels, nutritional status, and reading/writing/phonological processing abilities, showed significant associations of two omega-6 to omega-3 ratios with reading measures. The analyses confirmed the results that had been found by Borasio and colleagues on a smaller sample [35], showing that high AA/ALA and LA/ALA ratios are associated with worse performances in reading speed, writing, and phonological processing accuracy skills. The correlations are also in line with those of a previous study by Cyhlarova and colleagues [5], showing positive associations between omega-3 blood concentrations and reading performance and negative correlations between omega-6, omega-6/omega-3 and LA/ALA ratios and reading.

Regarding nutritional status, the Kcal omega-6 to total Kcal ratio was found to be negatively associated with reading measures: this is probably due to a very high omega-6 intake, characteristic of the typical western diet, and the lower intake of omega-3 PUFAs, which have been shown to be very important during child development [83,84]. Reading performance was expected to correlate with the amount of Kcal or carbohydrates. However, after Bonferroni correction, the association of reading speed with Kcal and carbohydrates did no longer reach significance (*p* = 0.037 and *p* = 0.039 respectively). This is interpreted as a limitation of sample size rather than a real lack of association. Therefore, we expect to find significant associations in larger samples of participants.

A more general comment concerns the dietary habits of the children. Considering the total number of children in the study, low adherence to the national dietary guidelines can be observed [1], e.g., concerning mean protein and lipid amounts from the diet. In fact, the national recommendations for this age range between 31–48 g/day for proteins and 20–35% for lipids. However, 95% of children declared to eat processed meat more than one time/week, to eat sweets almost once a day (recommendation for both: ≤once weekly) and to consume cheese more than the two times/week recommended. Moreover, about half of the children declared to consume vegetables/fruits almost two times/day (recommendation: five portions a day) and most of them declared not to eat nuts/seeds (recommendation: three times weekly). It is clear from this information that the sample is very far from national guidelines and the Mediterranean diet. Consequently, it is not surprising that a low omega-3 intake and a high omega-6/omega-3 ratio were observed.

Considering possible associations between dietary intake and PUFA levels in the blood, correlation analysis showed significant associations between the Kcal omega-6 to total Kcal ratio and the two AA/ALA and LA/ALA ratios. Higher levels of omega-6 dietary intake—with respect to the whole quantity of Kcal—were associated with higher levels of omega-6 to omega-3 in the blood, confirming expectations.

The main hypotheses of the present study, however, concerned the results of mediation and moderation models and predicted that the relationships among the three levels under investigation (dietary intake, blood PUFAs, and neuropsychological functions) would be more complex than simple linear, direct relationships. Indeed, such expectations were confirmed by the results of the study.

First of all, mediation analysis on the effects of nutritional intake on learning measures showed the indirect effect of dietary intake on reading speed mediated by the AA/ALA ratios in the blood. LA/ALA, by contrast, does not appear to be a significant mediator of the effects of Kcal omega-6 to total on reading speed, and the direct effect of both dietary intake and the LA/ALA ratio itself seem to be more relevant. Indeed, previous literature showed that children and adolescents may be particularly vulnerable to the nutritional effects of breakfast on brain activity and associated cognitive and academic outcomes [83]. This is due to the fact that children have a higher metabolic rate of glucose utilization and higher average cerebral blood flow and oxygen utilization as compared with adults. Moreover, they have a higher sleep demand, resulting in long overnight fasting periods, which can deplete glycogen stores overnight. Breakfast may, thus, provide a supply of energy to modulate the short-term metabolic responses to fasting conditions. It may also provide long-term benefits, such as an improved nutrient balance and distribution which may positively affect cognitive processes [84]. Further studies and a larger sample of children could allow more in-depth analyses to be conducted and clearer interpretations of such relationships to be formulated.

Moderation models, with learning performances as moderators in the relationship between dietary intake of fatty acids and blood PUFA levels, showed that the effects of nutritional intake on both considered blood level ratios (AA/ALA and LA/ALA) differ at different levels of reading speed performance. Thus, reading abilities appear to affect the relationship between the omega-6-derived calories to total amount of calories ratio and the relative balance of omega-6/omega-3 in the blood. This result is particularly interesting because it suggests that the presence of PUFAs in the blood and their ratios are not a simple consequence of the amount of PUFAs in the children’s diet. On the contrary, factors related to individual characteristics seem to play an important role. While this is not surprising as far as individual absorption capacity is involved—as shown in previous research (Mulder et al. [49]), it is noteworthy that such individual factors are linked to the children’s learning status. In particular, children with lower school learning abilities appear to be less able to process the dietary intake of PUFAs at the blood level. Constitutional and genetic vulnerability factors that form the basis for different levels of learning skills could be one explanation for this variability. Genetic transmission of reading disorders is well established and involves several genes [85], which may well have effects on metabolic variation. An association emerged between genetic vulnerability and DD [86,87] so that genes could explain 60% of the variability in reading attainment between subjects [88]. Nowadays, at least nine dyslexia risk loci termed DYX1–DYX9 on eight different chromosomes have been mapped and the involvement of several genes has been reported [85]. Other genes have been recently associated with reading (dis)ability [89]. The interaction of several genes, together with the impact of adverse environmental factors (e.g., immunological attack and poor nutrition) may compromise neuronal development, as it has been suggested for the magnocellular system, i.e., the part of the visual system specialized in the processing of low-spatial-frequency and high-temporal-frequency stimuli [87]. Initial evidence has also been provided suggesting that the presence of certain genetic variants may influence the expression of some of the DD-candidate genes [85]. As described above, LA/ALA has been suggested to act in a similar way, modulating the expression of genes and thus their overt effects on the organism and brain [86] PUFAs may influence gene expression profiles through several different mechanisms. These may be direct mechanisms, such as binding to nuclear transcription factors, or indirect mechanisms linked to the production of lipid mediators such as inositol triphosphate (IP_3_) and diacylglycerol (DAG), modulating cellular functions [90]. The effects of ALA seem to involve immune processes, as well as functions related to the maintenance and repair of DNA and RNA [86]. Some studies indeed suggest that dietary omega-3 LCPUFA supplementation may affect gene expression through epigenetic changes [36], i.e., through non-coding RNAs, histone modifications, and DNA methylation. In adult humans, differences in DNA methylation have been identified between people with high and low omega-3 PUFA intakes. As for pediatric clinical trials, due to the small population and gaps in follow-up time, further studies are needed to provide more answers on the influence of omega-3 LCPUFAs on gene expression and modulation, particularly regarding the benefits for newborns and their future health [91].

A different possibility is that a low LA/ALA ratio may have potential benefits for the control of inflammatory disease pathogenesis, whereas higher LA/ALA ratios have a pro-inflammatory effect [92].

As to the AA/ALA ratio, although very little is found in the literature, it appears to determine the ratio between two molecules competing to convert to bioactive eicosanoids. The AA produces prostaglandins and thromboxanes, as well as leukotrienes, which have strong proinflammatory and vasoconstriction effects. By contrast, eicosanoids derived from omega-3 PUFA are considered to be antithrombotic, anti-inflammatory, and vasodilating. Therefore, a well-balanced AA/ALA ratio may be crucial in regulating the production of inflammatory mediators [93]. Both DHA and AA were shown to influence neurogenesis through their effects on neural stem cells (NSC) proliferation and differentiation. Reduced omega-3 PUFAs in early life interferes with the migration of neuronal cells in the brain [94]. The role of AA in neurodevelopmental disorders could further be linked to gliogenesis. Indeed, LA dietary supply seems to promote gliogenesis through AA-derivatives [95].

In general, the results of the present study highlight the crucial role of shorter-chain (LA and ALA) precursors of LCPUFAs in the relationship with reading abilities. Since this relationship seems to be even stronger than that found for DHA and EPA levels, it can be hypothesized that shorter-chain PUFAs exert their own role beyond their function in the synthesis or absorption of LCPUFAs. Although several studies showed a significant association of blood levels of EPA and DHA with reading and writing measures [5], these PUFAs of the omega-3 group did not show very large effects in our sample. Nonetheless, it should be considered that reduced sample size and correction for multiple testing may have obscured effects that are indeed present (the correlation of DHA and AA/EPA with reading time is not negligible, even if not significant). Moreover, different results might have emerged considering brain levels (instead of blood levels) of fatty acids [96] or considering typically developing and reading-impaired individuals separately, which our current sample size does not permit but is envisaged for future studies. A final consideration concerns the role of FADS. In fact, although the conversion rate of ALA to EPA and to DHA is very low in humans [97], the infant’s capacity for endogenous synthesis of AA and DHA from their precursors could depend on genetic variation in the production of these enzymes [98]. A hypothetical link with other genetic variations possibly related to reading ability cannot be excluded at present, even if no data are available to support this possibility. More generally, a relatively high LA intake (and a consequent higher presence of AA) in infants could reduce omega-3 synthesis, resulting in a lower DHA status. It should be considered, yet, that the possible effects of imbalances in omega-3/omega-6 intakes on circulating DHA levels are complex and not limited to the LA/ALA ratio, and that a relevant role is played by the total dietary PUFAs intake [99].

Altogether, the results from the present study provide a few answers to the initial experimental questions but leave many issues to be further investigated and clarified.

Among other things, future studies should clarify the role of the intake of LCPUFAs in pregnant mothers, in the perspective of transgenerational transmission of health, possibly extending to the dietary style of both the mother and the father in the pre-conception phase [38].

It should be noted that children with typical development and children with DD were considered as a whole group in the present study since their reading and writing skills were found to be distributed along a continuum from good to impaired abilities. However, the underlying mechanisms of learning ability might be different in the two groups. For the same reason, fatty acid profiles and the association with PUFA dietary intake could be different amongst children with or without the neurodevelopmental disorder, as found in previous research [8,52,54]. Unfortunately, the sample size of the two sub-groups of children did not allow us to run separate analyses, and this could be considered a limitation of the study. Even if the relatively small sample size of this study calls for caution and prevents strong claims, the results provide new, interesting data on the relationships between intake and blood levels of PUFA and learning abilities (reading and writing skills). More specifically, the results of the mediation and moderation models (to be confirmed in larger samples) suggest that both dietary and constitutional factors contribute to modulating the fatty acids profile along the reading ability continuum.

## 5. Conclusions

The present study has confirmed the associations between PUFA levels in the blood, dietary intake, and learning abilities., regardless of the distinction between children with and without a diagnosis of DD. Children’s performances in reading, writing, and phonological awareness appear to be sensitive to the effects of favorable conditions related to a higher intake of omega-3 PUFAs reflected in the higher omega-3 blood levels. The relationship between dietary intake and PUFA levels in the blood appears to be moderated by possibly genetic/constitutional factors as expressed by variation in the reading and writing performance scores with respect to the normative population. The present results provide evidence of the role of dietary intake and PUFA levels in creating the basis for the development of reading and writing skills and encourage consideration of such factors in the design of prevention and intervention programs for children with learning disorders.

## Figures and Tables

**Figure 1 biomolecules-13-00368-f001:**
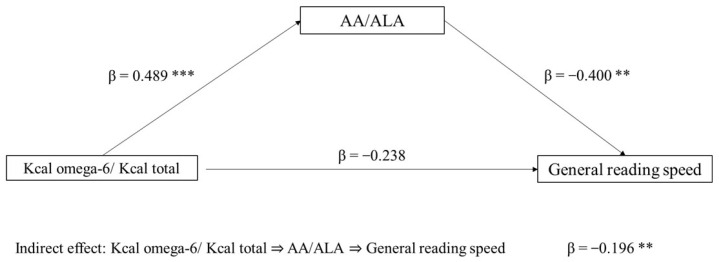
Mediation model with general reading speed (z-score) as the dependent variable, Kcal omega-6/Kcal total as the independent variable, and AA/ALA ratio as mediator (** *p* < 0.05, *** *p* < 0.001, one-tailed). The figure shows direct and indirect (component) effects. Betas are completely standardized effect sizes.

**Figure 2 biomolecules-13-00368-f002:**
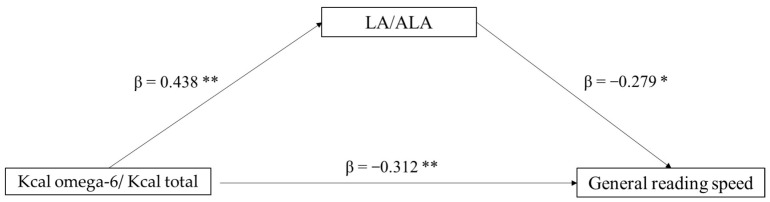
Mediation model with general reading speed (z-score) as the dependent variable, Kcal omega-6/Kcal total as the independent variable, and LA/ALA ratio as mediator (* *p* < 0.10, ** *p* < 0.05, one-tailed). The figure shows direct and indirect (component) effects. Betas are completely standardized effect sizes.

**Table 1 biomolecules-13-00368-t001:** Participants’ characteristics, performance profiles in reading/writing/phonological awareness, and PUFA levels. Learning measures are expressed as z-scores while PUFA levels are reported as percentages or ratios to fatty acids raw scores.

	Mean (SD)	Range
Age	10.68 (1.39)	8.17–13.58
IQ	102.64 (13.31)	83–128
General reading accuracy	−1.04 (1.60)	−5.54–1.17
General reading speed	−1.91 (2.50)	−9.44–1.23
General writing accuracy	−1.28 (2.36)	−7.75–1.17
Phonological processing accuracy	45.74 (4.16)	35–52
Total omega-3	2.75 (0.70)	1.56–5.22
Omega-6/omega-3	12.39 (3.30)	5.75–22.94
DHA	1.75 (0.48)	0.90–2.97
AA/EPA	32.11 (18.26)	9.54–86.80
AA/ALA	73.88 (43.80)	23.62–244.50
LA/ALA	173.00 (89.44)	67.76–483.00

**Table 2 biomolecules-13-00368-t002:** Children’s nutritional status: average intake for energy and macronutrients and for SFA, PUFA, MFA, and ratios.

	Mean (SD)	Range
Energy (kcal)	2047.60 (409.93)	1434–3032
Carbohydrates (g)	290.44 (76.32)	143.5–495
Lipids (g)	70.86 (13.48)	44.7–98.3
Total PUFA (g)	8.46 (1.93)	4.75–13.95
kcal n-6/kcal tot (%)	3.13 (0.57)	2.25–4.69
kcal n-3/kcal tot (%)	0.51 (0.14)	0.00–1.04

**Table 3 biomolecules-13-00368-t003:** Partial correlations (controlling for age) between PUFA levels and reading/writing/phonological processing measures (*n* = 42, alpha = 0.017). Reported values are Pearson’s correlation index r/*p*-value. In bold, significant correlations.

	General Reading Time	General Reading Errors	General Writing Errors	Phonological Awareness Accuracy
**Total omega-3**	−0.147/0.358	−0.149/0.352	−0.179/0.263	0.140/0.383
**Omega-6/omega-3 ratio**	0.071/0.660	0.255/0.108	0.199/0.213	−0/195/0.222
**DHA**	−0.137/0.392	−0.089/0.580	−0.137/0.393	0.076/0.635
**AA/EPA ratio**	−0.007/0.996	0.219/0.170	0.117/0.468	−0.181/0.256
**AA/ALA ratio**	**0.617/<0.001**	0.154/0.335	0.354/0.023	−0/355/0.023
**LA/ALA ratio**	**0.539/<0.001**	0.192/0.228	0.358/0.021	−0.329/0.036

**Table 4 biomolecules-13-00368-t004:** Partial correlations (controlling for age) between nutritional information and reading/writing/phonological processing measures (*n* = 42, alpha = 0.017). Reported values are Pearson’s correlation index r/*p*-value. In bold, significant correlations.

	General Reading Time	General Reading Errors	General Writing Errors	Phonological Awareness Accuracy
**Energy (kcal)**	−0.326/0.037	−0.374/0.016	−0.229/0.149	0.322/0.040
**Carbohydrates (g)**	−0.324/0.039	**−0.393/0.011**	−0/172/0.282	0.364/0.019
**Lipids (g)**	−0.205/0.199	−0.203/0.202	−0.267/0.092	0.135/0.401/
**Total PUFA (g)**	−0.020/0.901	−0.071/0.659	0.078/0.626	−0.185/0.247
**kcal n-6/kcal tot (%)**	**0.476/0.002**	0.282/0.074	**0.443/0.004**	**−0.640/<0.001**
**kcal n-3/kcal tot (%)**	0.020/0.900	0.186/0.244	−0.111/0.488	−0.321/0.041

**Table 5 biomolecules-13-00368-t005:** Partial correlations (controlling for age) between PUFA levels and nutritional information (*n* = 42, alpha = 0.006). Reported values are Pearson’s correlation index r/*p*-value. In bold, significant correlations.

	Total Omega-3	Omega-6/ Omega-3 Ratio	DHA	AA/EPA Ratio	AA/ALA Ratio	LA/ALA Ratio
**Energy (kcal)**	0.173/0.279	−0.190/0.234	0.123/0.444	−0.248/0.188	−0.223/0.162	−0.178/0.265
**Carbohydrates (g)**	0.129/0.421	−0.196/0.218	0.084/0.601	−0.226/0.155	−0.233/0.143	−0.220/0.166
**Lipids (g)**	0.146/0.364	−0.56/0.729	0.108/0.500	−0.192/0.228	−0.077/0.634	0.007/0.967
**Total PUFA (g)**	0.173/0.280	−0.174/0.276	0.095/0.555	−0.281/0.075	0.091/0.572	0.129/0.420
**kcal n-6/kcal tot (%)**	−0.133/0.483	0.062/0.701	−0.152/0.341	−0.057/0.721	**0.515/0.001**	**0.503/0.001**
**kcal n-3/kcal tot (%)**	0.396/0.010	−0.307/0.051	0.388/0.012	−0.024/0.880	−0.068/0.673	−0.043/0.791

**Table 6 biomolecules-13-00368-t006:** Moderation models: analyses of the effect of learning measures (z-scores) in the associations of Kcal omega-6/Kcal total (predictor) with AA/ALA and LA/ALA ratios (dependent variables).

Predictor	Moderator	Dependent Variable	Estimate	SE	Lower	Upper	*Z*	*p*
Kcal omega-6/Kcal total	General reading speed	AA/ALA	−7.86	2.43	−12.63	−3.091	−3.23	0.001
		LA/ALA	−17.58	5.31	−27.99	−7.17	−3.31	<0.001

## Data Availability

Data are not available due to the ongoing main project n. 677 and to restrictions set by the Ethical Committee on data sharing. Nonetheless, data can be made available upon written request to the corresponding author, pending appropriate agreements and approval by the institutional review board.

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
