# Peer review of "Associations between Dietary Intake, Blood Levels of Omega-3 and Omega-6 Fatty Acids and Reading Abilities in Children"

_biomolecules, 2023, doi:10.3390/biom13020368_

Round 1

Reviewer 1 Report

This is a potentially very interesting paper that seems to confirm previous results but by using a very large number of pretty arbitrary biochemical measurements.   Moreover none of the dietary measures seem to have been analysed..

Although I am not a statistician, I am also bothered that so many correlations were computed from only 42 subjects' measurements and significances claimed without any apparent attempt to allow for this large number of computations, in particular where only a 1 tailed significance test is used.  A simple Bonferoni approach would suggest that very few of them would have survived this harsh razor.     

In addition many of the variables investigated, such as n-6 calories/total cals seem very arbitrary and should be justified better.  Also no mention of omega 3 index or of DHA or EPA alone.  Thus the correlations that were not found to be significant should be pointed out, eg presumably n-3 index, EPA, DHA.  Finally I am highly sceptical of mediation v. moderation analysis, especially with such a limited number of subjects; they must be better described and defended.

Reviewer 2 Report

I have indicated my comments (in red font color) in the attached pdf-file version of the manuscript.

In general,the English style should be improved. Some sentences are very hard to digest. 

Round 2

Reviewer 1 Report

This paper has been improved greatly and could be published. However, since Harris's omega 3 index is widely used I think you should justify, around line 247 and in the discussion, why you didn't use it. Also one of the most interesting outcomes in the whole paper is that short chain PUFA ratios seem to affect reading etc.  more than AA, DHA and EPA.  This merits more emphasis and discussion.

Reviewer 2 Report

In general I enjoyed reading the revised version. It is sounder than the initial submission (especially the results and discussion section). However, for the sake of clarity I made some minor comments/corrections, highlighted in green. More specifically: in the Title; in lines 345-346; Tables 3 to 5 and line 794.
